Reconstitution of the complete rupture in musculotendinous junction using skeletal muscle-derived multipotent stem cell sheet-pellets as a “bio-bond”

Hashimoto Hiroyuki 1 2
Tamaki Tetsuro tamaki@is.icc.u-tokai.ac.jp 2 3
Hirata Maki 1 2
Uchiyama Yoshiyasu 1 2
Sato Masato 1
Mochida Joji 1
1 Department of Orthopaedic, Tokai University School of Medicine , Isehara , Japan
2 Muscle Physiology and Cell Biology Unit, Tokai University School of Medicine , Isehara , Japan
3 Department of Human Structure and Function, Tokai University School of Medicine , Isehara , Japan
Abdala Virginia
Electronic publication date: 2016 Jul 19
Publication date: 2016
Volume: 4
Electronic Location ID: e2231
Received 2016 Jan 23; Accepted 2016 Jun 16
Copyright: ©2016 Hashimoto et al.
Copyright year: 2016
Copyright holder: Hashimoto et al.
License: This is an open access article distributed under the terms of the Creative Commons Attribution License, which permits unrestricted use, distribution, reproduction and adaptation in any medium and for any purpose provided that it is properly attributed. For attribution, the original author(s), title, publication source (PeerJ) and either DOI or URL of the article must be cited.
License URL: https://creativecommons.org/licenses/by/4.0/

Keywords: Muscle regeneration, Nerve reconstitution, Tendon regeneration, Stem cell therapy, Vascular reconstitution, Musculotendinous junction, Muscle contraction, Severe muscle injury

Funding: Tokai University School of Medicine, Project Research (TT) This work was partially supported by the grant 2013–2014 Tokai University School of Medicine, Project Research (TT). The funders had no role in study design, data collection and analysis, decision to publish, or preparation of the manuscript.

==============================
Background. Significant and/or complete rupture in the musculotendinous junction (MTJ) is a challenging lesion to treat because of the lack of reliable suture methods. Skeletal muscle-derived multipotent stem cell (Sk-MSC) sheet-pellets, which are able to reconstitute peripheral nerve and muscular/vascular tissues with robust connective tissue networks, have been applied as a “bio-bond”.

Methods. Sk-MSC sheet-pellets, derived from GFP transgenic-mice after 7 days of expansion culture, were detached with EDTA to maintain cell–cell connections. A completely ruptured MTJ model was prepared in the right tibialis anterior (TA) of the recipient mice, and was covered with sheet-pellets. The left side was preserved as a contralateral control. The control group received the same amount of the cell-free medium. The sheet-pellet transplantation (SP) group was further divided into two groups; as the short term (4–8 weeks) and long term (14–18 weeks) recovery group. At each time point after transplantation, tetanic tension output was measured through the electrical stimulation of the sciatic nerve. The behavior of engrafted GFP+ tissues and cells was analyzed by fluorescence immunohistochemistry.

Results. The SP short term recovery group showed average 64% recovery of muscle mass, and 36% recovery of tetanic tension output relative to the contralateral side. Then, the SP long term recovery group showed increased recovery of average muscle mass (77%) and tetanic tension output (49%). However, the control group showed no recovery of continuity between muscle and tendon, and demonstrated increased muscle atrophy, with coalescence to the tibia during 4–8 weeks after operation. Histological evidence also supported the above functional recovery of SP group. Engrafted Sk-MSCs primarily formed the connective tissues and muscle fibers, including nerve-vascular networks, and bridged the ruptured tendon–muscle fiber units, with differentiation into skeletal muscle cells, Schwann cells, vascular smooth muscle, and endothelial cells.

Discussion. This bridging capacity between tendon and muscle fibers of the Sk-MSC sheet-pellet, as a “bio-bond,” represents a possible treatment for various MTJ ruptures following surgery.

Introduction

Skeletal muscles represent muscle–tendon complexes attached to the bone. However, due to their roles in the protection of the body and the force generation at the body-movements, injuries invariably occur during various activities or as a result of accidents. Muscle injuries can be classified as ruptures, tears, and lacerations, typically caused by external hard compression (contusion) or excessive stretching forces, and they are categorized into 3 grades of severity, as follows: Grade-I (mild) injury affects only a limited number of fibers in the muscle, and the strength does not decrease in the full active and passive range of motions, with pain and tenderness being delayed until the next day; Grade-II (moderate) injury, where nearly half of muscle fibers are torn, and acute and significant pain is accompanied by swelling and a minor decrease in muscle strength; Grade-III (severe) injury, with the complete rupture of the muscle, where the injured muscle is torn into 2 parts, together with severe swelling and pain, and a total loss of function. The injuries can be broadly divided according to their location in the muscle belly, musculotendinous junction (MTJ), and tendon tear (Chan et al., 2012; ElMaraghy & Devereaux, 2012). Generally, for Grade-I and -II injuries, conservative treatments are usually applied, but in the case of Grade-III injuries, surgical intervention is often considered (Kragh et al., 2005c; Oliva et al., 2013; Rawson, Cartmell & Wong, 2013). Several suturing techniques have been reported for the treatment of complete tendon rupture (Hirpara et al., 2007; Maquirriain, 2011; Merolla et al., 2009; Rawson, Cartmell & Wong, 2013; Yildirim et al., 2006), but there is a lack of reliable suture methods for the ruptures that involve the muscle belly or MTJ (Faibisoff & Daniel, 1981; Kragh et al., 2005c; Oliva et al., 2013; Phillips & Heggers, 1988). Complete rupture at the MTJ is particularly difficult, whereas suture of the muscle belly has been previously attempted (Kragh Jr et al., 2005a; Kragh Jr et al., 2005b; Kragh Jr et al., 2005c). The maintenance of the continuity in muscle–tendon unit is important, and strengthening of the adhesion properties is absolutely imperative, as there is a risk of repeated rupture after both surgical and/or non-surgical treatments (Kircher et al., 2010; Young et al., 2014).

Furthermore, tearing of the muscle–tendon unit can damage small blood vessels and nerves, generally causing local bleeding, pain, and/or paralysis. Therefore, the early re-establishment of peripheral nerve and blood vessels is important for the muscle repair process, in order to maintain the supply of O2 and other nutrients, and the removal of CO2 and other waste products (Ackermann, Ahmed & Kreicbergs, 2002; Ackermann et al., 2003; Nishimori et al., 2012). Additionally, it has been suggested that the repair of tendon ruptures can be stimulated by a single application of several growth factors, including platelet-derived growth factor (PDGF) (Hildebrand et al., 1998), transforming growth factor (TGF)-β (Kashiwagi et al., 2004), insulin-like growth factor (IGF)-1 (Kurtz et al., 1999), basic-fibroblast growth factor (bFGF) (Chan et al., 2000), and vascular endothelial growth factor (VEGF) (Zhang et al., 2003). Synchronized supply of these factors is considered beneficial for the reconstruction of the muscle–tendon unit. Therefore, the application of an adhesive able to connect muscles to tendons may be a good treatment strategy for MTJ injury. Several scaffolds have been applied in the tendon healing treatments, and recent tissue-engineering investigations have shown that cell-scaffold constructs can improve the healing of tendon defects, compared with scaffolds alone (Ouyang et al., 2002; Ouyang et al., 2003; Young et al., 1998). Bone marrow-derived mesenchymal stem cells are most frequently applied as adjuvant cells, and their favorable healing effects have been reported (Chong et al., 2007; Harris et al., 2004; Lu et al., 2016; Ouyang, Goh & Lee, 2004; Vieira et al., 2014), while the behavior of the transplanted cells, in terms of engraftment and differentiation, is poorly understood. An adipose-derived stem cell was also applied to the tendon repair, and a significant increase in tensile strength associate with the differentiation into tenocytes and endothelial cells were reported (Uysal & Mizuno, 2011). Similarly, skin-derived tenocyte-like cells was also used for the treatment of patellar tendinopathy, and greater improvement in pain and function was suggested (Clarke et al., 2011). However, these were applied to the tendon repair itself, thus, the effects for the MTJ rupture is still not clear.

We have determined that skeletal muscle-derived multipotent stem cells (Sk-MSCs) are capable of synchronized reconstitution of muscle-nerve-blood vessel unit and cellular differentiation into skeletal muscle cells, Schwann cells, perineurial/endoneurial cells, pericytes, vascular smooth muscle cells, and endothelial cells (Tamaki et al., 2007a; Tamaki et al., 2005). Recently, we developed a 3D gel-patch tissue reconstitution system using Sk-MSC sheet-pellets, which are able to preferentially reconstitute peripheral nerve and vascular tissues with robust connective tissue networks (Tamaki et al., 2013). Sk-MSC sheet-pellets also expressed various neurotropic/neurotrophic and vasculogenic factor mRNAs before and after transplantation (Soeda et al., 2013; Tamaki et al., 2013). These properties of Sk-MSCs and their sheet-pellets are considered to be beneficial for the reconstitution of muscle–tendon units, including their nerve-blood vessel networks. In this study, we developed a complete rupture model for MTJ in tibialis anterior (TA) muscle of mice, and applied Sk-MSC sheet-pellets as a “bio-bond”-like substance. Functional recovery, which was evaluated by the electrical stimulation-induced muscle contractions via the sciatic nerve, was measured and the behavior of engrafted cells was analyzed immunohistochemically. The putative paracrine capacity of growth factors in the sheet-pellets, relating to skeletal muscle, nerve, and vascular regeneration, was confirmed by RT-PCR and protein array.

Materials and Methods

Animals

Green fluorescent protein transgenic mice (GFP-Tg mice; C57BL/6 TgN[act EGFP]Osb Y01, provided by Dr. M Okabe, Osaka University, Osaka, Japan) (Okabe et al., 1997) were used as donor mice for the cell transplantation experiments (male, 4–8 week old, n = 6), and wild-type mice (C57BL/6N) were used as recipients (male, 8–12 week old, n = 26). All experimental procedures were approved by the Tokai University School of Medicine Committee on Animal Care and Use (153015).

Figure 1 Macroscopic and schematic images of procedures used in MTJ complete rupture model preparation and sheet-pellet transplantation.

(A) An overview of the sheet-pellet preparation. (B) A gel-like state of sheet-pellet being picked up with forceps. Photographs of a left-TA muscle were taken. (C) Step 1; (D) Step 2; and (E) Step 4. Dotted circle in (B) shows peeling of fibers. White arrows in (C) show partially sutured skin. Bars = 1 mm.

Cell purification and preparation of stem cell sheet-pellets

Sk-MSC sheet-pellets (Tamaki et al., 2013) were investigated for their effects on the regeneration of completely ruptured MTJ. Sheet-pellets generally showed gel-like characteristics (Fig. 1A), and they could be lifted using forceps (Fig. 1B). The thigh and lower leg muscles (tibialis anterior, extensor digitorum longus, soleus, plantaris, gastrocnemius, and quadriceps femoris) of GFP-Tg mice were removed and used in subsequent experiments. Muscle sampling was performed under an overdose of pentobarbital (60 mg/kg, Schering–Plough, combined with butorphanol tartrate 2 mg/kg, Meiji Seika, Tokyo, Japan, i.p.). Average total muscle mass removed during the procedure was 512 ± 67 mg/GFP-Tg mouse (mean ± SE). Muscles were not minced, and were subsequently treated with 0.1% collagenase type IA (Sigma–Aldrich, Tokyo, Japan) in Dulbecco’s modified Eagle’s medium (DMEM, Wako, Osaka, Japan) containing 7.5% fetal calf serum (FCS, Equitech Bio, TX, USA) with gentle agitation for 30 min at 37 °C. Following a short digestion, whole muscles were divided into fiber-bundles, which were washed with culture medium (Iscove’s modified Dulbecco’s medium; IMDM, Wako, Osaka, Japan) containing 10% FCS, and cultured in IMDM/20% FCS with 100 units/ml penicillin G, 100 µg/ml streptomycin sulfate (Wako, Osaka, Japan), 10 µg/ml gentamycin sulfate (Schering–Plough, Osaka, Japan), and 0.1 mM β-mercaptoethanol (Wako, Osaka, Japan) for 3 days. Cultured fiber-bundles associated with expanded cells were treated with trypsin-EDTA (0.05% trypsin, 0.53 mM EDTA; Life Technologies, Tokyo, Japan), in order to dissociate individual cells. Single-cell suspension was filtered through 70-, 40- and 20-µm nylon filters in order to remove muscle fibers and other debris, and, after washing, isolated cells were re-cultured in IMDM/20% FCS for 2–3 days, until they reached confluence. In total, cells were expanded in culture for 5–6 days, which represents a reduced culture time compared with previous studies (Soeda et al., 2013; Tamaki et al., 2013), thus preserving the myogenic potential of sheet-pellets. After they reached confluence, the cells were gently detached from culture dishes using 2 mM EDTA solution. This step mainly affected Ca2+-dependent cell adhesion (e.g., cadherins), while cell-to-cell contact was maintained, sheet-like cell aggregations were collected and centrifuged, and stem cell sheet-pellets were obtained. Total sheet-pellet mass was 120 ± 17 mg/mouse (mean ± SE), which means that over 100 mg of sheet-pellets were obtained from about 500 mg of skeletal muscle tissue. Throughout cell isolation, 7.5–10% FCS was added to the collagenase and washing solutions in order to minimize contaminating protease activity and to protect isolated cells as much as possible.

RT-PCR and protein analysis of the sheet-pellets

Quality, differentiation potential and relative expressions of cytokines of the cells forming sheet-pellets was confirmed by RT-PCR and antibody array analysis immediately before the transplantation. Cells composed sheet-pellet was prepared for the RT-PCR, and their supernatant was prepared for antibody array analysis. For the RT-PCR analysis, specific primers and the analyzed materials are summarized in Table S1. Cells were lysed and total RNA was purified using a QIAGEN RNeasy Micro Kit (Hilden, Germany). First-strand cDNA synthesis was performed with Invitrogen SuperScript III system using dT30-containing primer (see Table S1), and specific PCR (35 cycles of 30 s at 94 °C, 30 s at 60–65 °C and 2 min at 72 °C) was performed in a 15-µl volume containing Ex-Taq buffer, 0.8 U of ExTaq-HS-polymerase, 0.7 µM specific sense and antisense primers, 0.2 mM dNTPs, and 0.5 µl of cDNA. Relative expression was normalized to the expression of a housekeeping control (HPRT). Details of this analysis were described previously (Tamaki et al., 2013).

Concurrently, several cytokines, which are related to the muscular and vascular regeneration, were also analyzed by antibody array kit (Proteome Profiler, ARY013, R & D Systems, Minneapolis, MN), as a protein level. Cell culture supernatant of sheet-pellet just before the transplantation was obtained after removal of particulates by centrifugation, and 500-µl of supernates was prepared for the analysis. Culture medium containing 20% FCS was also prepared for the same analysis in order to check the background effects. The relative expression levels of several cytokines; such as interleukin-6 (IL-6), insulin-like growth factor-1 and 2 (IGF-1 and 2) and their related proteins IGF binding protein-1, 2, 5, and 6 (IGFBP-1, 2, 5 and 6), which are the critical regulator of myogenesis; a regulator/enhancer of macrophage such as chemokine (C–C motif) ligand 2 (MCP-1) and colony stimulating factor 1 (M-CSF); a regulator/activator of wide range of cell types such as fibroblast growth factor-21 (FGF-21) and tissue inhibitor of metalloproteinase 1 (TIMP-1); and vascular endothelial growth factor (VEGF) were determined.

Complete rupture of MTJ model and application of sheet-pellets

In order to prepare the experimental complete rupture model, we manually detached muscle fibers from the distal tendons of the right tibialis anterior (TA) muscles of recipient mice (n = 19). All surgical preparations were performed under the inhalation anesthesia (Isoflurane; Abbot, Osaka, Japan). A summary of the procedure is shown in Fig. 1. First, TA muscle was exposed (C, step 1), and muscle fibers were subsequently detached from the distal tendon and peeled off using cotton swabs for one half of the entire TA length (D, step 2), and everted necrotic fiber portions were removed (step 3). The average removed muscle mass was 21.5 ± 1.8 g, and this represented about 40% of total muscle mass. Sheet-pellets were then adhered to the open region of completely ruptured MTJ (Fig. 1E, step 4; Sheet-pellet, SP group, n = 14), and skin was sutured (E, white arrows). The left side was preserved as a contralateral control. The non-transplanted control group (C group, n = 10) underwent the same surgery, and the same amount of cell-free culture medium was administered. The mice were allowed full freedom of movement after surgery.

Functional assessment of the regenerated TA muscles

After the transplantation, the animals were divided into two groups as short-term (4–8 weeks) and long-term (14–18 weeks) recoveries, and prepared for the functional assessment. The first assessment was begun at 4 weeks using each one mouse in SP and C group, and tetanic tension outputs of regenerated TA muscles were measured in both left (non-operated control side) and right (operated side) legs. The tetanic tension output was considered as the total functional recovery of the operated TA muscle, and recovery ratio was determined based on the contralateral non-operated control side. Following assessments were performed every week (0.5–1.5 week interval) during each group terms using one by one mouse, then, SP and C groups were compared. Measurements were performed in situ under inhalation anesthesia (Isoflurane; Abbot, Osaka, Japan), and body (rectal) temperature was maintained at 36 ± 1 °C with a radiant heat light throughout the measurement. Details of this measurement were as described previously (Tamaki et al., 2005). Briefly, the distal tendon of TA muscle and sciatic nerve (about 10 mm) on both sides were carefully exposed, and tissues were coated with mineral oil to prevent tissue drying and to minimize electrical noise interference. A bipolar silver (Ag/Ag) electrode (inter-electrode distance: 2 mm) was placed under the sciatic nerve. A stainless steel hook was attached to the distal tendon of each TA muscle using a silk ligature. The animal was transferred to a custom-made operating table that allowed stabilization of the head and limbs in a supine position using surgical tape. A stainless steel hook was attached to a force-distance transducer (FD-Pickup, TB-611T; Nihon Kohden, Tokyo, Japan) connected to the carrier amplifier (AP-621G; Nihon Kohden). This enabled the measurements of the muscle contraction force and its distance to be conducted. A bipolar silver electrode (inter-electrode distance: 5 mm/1 mm diameter) was attached to the surface of the reference muscle as well, in order to obtain an evoked electrical myogram, as a confirmation of stable muscle contractions. We have taken care to avoid interference of the reference muscle and nerves with the normal blood supply. Afterward, twitches were elicited by single pulse (1 ms duration, 0.5 Hz) electrical stimulation via the sciatic nerve, at a voltage above the threshold for a maximum response (1.5–4.0 V). Subsequently, peak tetanic tension was determined using stimulation frequencies of 10, 20, 40, 60, 80, 100, 120, and 140 Hz of 0.5 s duration at 15 s intervals. The frequency that produced the highest tetanic tension was considered the optimal stimulation for tetanus. All mechanical and electrical measurements were recorded on a Linearcorder (Mark VII, WR3101; Graphtec, Tokyo, Japan) as analog data.

Macroscopic observation and immunostaining

Following the functional measurements above, recipient mice (including the animals used for morphological analysis only, 5–10 weeks after transplantation, n = 5) were given an overdose of pentobarbital (60 mg/kg, combined with butorphanol tartrate 2 mg/kg, i.p.), and the engraftment of donor-derived GFP+ cells into the damaged portion of TA muscle was confirmed by fluorescence stereomicroscopy (SZX12; Olympus, Tokyo, Japan, Fig. 4). Recipient mice were perfused with warm 0.01 M phosphate-buffered saline (PBS, Wako, Osaka, Japan) through the left ventricle, followed by fixation with 4% paraformaldehyde/0.1 M phosphate buffer (4% PFA/PB, Wako, Osaka, Japan). Muscles were removed and fixed overnight in 4% PFA/PB, washed with graded sucrose (0–25%, Wako, Osaka, Japan)/0.01 M PBS series, and quick frozen with isopentane (Wako, Osaka, Japan) pre-cooled by liquid nitrogen, followed by storage at −80 °C. Subsequently, 7 µm cross-sections were obtained. Skeletal muscle fibers were stained with anti-skeletal muscle actin (αSkMA; dilution, 1:200; incubation, room temperature for 2 h; Abcam, Cambridge, UK). Nerve fiber localization (axons) was detected by rabbit polyclonal anti-Neurofilament 200 (N-200, dilution, 1:1000; incubation, room temperature for 1 h; Sigma, St. Louis, MO, USA). Schwann cells were detected using anti-p75 (rabbit polyclonal, 1:400, 4 °C overnight; CST, Boston, MA, USA). Blood vessels were detected with rat anti-mouse CD31 (1:500, 4 °C overnight; BD Pharmingen, San Diego, CA, USA) monoclonal antibody, which is a known vascular endothelial cell marker, and mouse monoclonal α-smooth muscle actin (αSMA, Cy3-conjugated; 1:1500; room temperature for 1 h; Sigma, St. Louis, MO, USA). Dystrophin formation in the skeletal muscle fibers was detected using goat anti-dystrophin polyclonal antibody (1:50, 4 °C, overnight; Santa Cruz Biotechnology, Dallas, TX, USA). Neuromuscular junctions were detected by α-bungarotoxin (Alexa Fluor 594 conjugated, 1:100, room temperature,1 h; Molecular Probes, Eugene, OR, USA). Reactions were visualized using Alexa Fluor-594-conjugated goat anti-rabbit and anti-rat antibodies (1:500, room temperature, 2 h; Molecular Probes, Eugene, OR, USA). Nuclei were counter-stained with DAPI (4,6-diamino-2-phenylindole).

Statistical analysis

Differences between two groups (short and long term recovery group) were tested using Student’s t test, and the significance level was set at p < 0.05. Values are expressed as mean ± SE.

Figure 2 RT-PCR analysis of Sk-MSC sheet-pellet and its expressions of cytokines immediately prior to transplantation.

(A) Expressions of myogenic, neurotrophic, and vasculogenic factor mRNAs was observed, confirming the quality of the sheet-pellet preparation. bp, base pair. (B) Several cytokines related to the muscle and vascular regeneration was also detected in the culture supernatant of the sheet-pellet, confirming the putative capacity of paracrine.

Results

Quality and therapeutic potential of Sk-MSC sheet-pellets

Quality, differentiation, and putative therapeutic potential of the sheet-pellet were first confirmed by RT-PCR and protein array analysis (Fig. 2). The expression of specific myogenic, neurotrophic, and vasculogenic factor mRNAs in the sheet-pellets immediately before the transplantation is shown in Fig. 2A. The Sk-MSC sheet-pellets showed expressions of various myogenic factors (MyoD, Myf5, Pax7, Myogenin, c-met, Mcad, MyH, Desmin, and IGF-1), neurotrophic factors (NGF, BDNF, GDNF, CNTF, LIF, Ninjurin, Galectin, Nestin, and Sox10) and vascular growth factors (VEGF, HGF, PDGF, TGF-β, EGF, and FGFb), except Pax3. These results agree with a previous report (Tamaki et al., 2013), which showed good quality of this method of sheet-pellet preparation.

In addition, relative increase in the expressions of proteins (cytokines) was also detected in the same sheet-pellet culture supernatant (Fig. 2B). An increase of myokine (Munoz-Canoves et al., 2013; Pedersen, 2012), which is a critical regulator of muscle regeneration, such as IL-6 and IGF-1 and 2 associate with their relating IGFBP-2, 3, 5 and 6, the chemokine, which is an up-regulator of monocyte/macrophage (MCP-1 and M-CSF) in the tissue regeneration (Pantsulaia et al., 2005; Shiba et al., 2007), and a regulator/activator of a wide range of cell types (FGF-21 and TIMP-1) (Mas et al., 2007; Wan, 2013) was detected, showing a paracrine capacity of the sheet-pellet.

Table 1 Functional assessment of the operated muscle.

Experimental group	Short-term (4–8 weeks) recovery group (n = 9)	Long-term (14–18 weeks) recovery group (n = 5)	
		Muscle mass (mg)	Tetanic tension N (1 × 102)	Muscle mass (mg)	Tetanic tension N (1 × 102)	
SP group (n = 14)	Op-side	31.1 ± 2.7	20.5 ± 3.5	39.9 ± 3.8	35.0 ± 7.8*	
Cont-side	49.4 ± 2.3	77.5 ± 5.1	51.1 ± 1.0	73.5 ± 9.7	
C group (n = 5)	Op-side	26.3 ± 1.4	Unmeasurable	Not done	Not done	
Cont-side	53.8 ± 1.6	79.7 ± 3.8	Not done	Not done	
Notes.

Values are expressed average ±SE.

* p < 0.05 4-weeks vs. 10 weeks tetanus.

Functional recovery

The results of functional assessment are summarized in Table 1, and group differences between the short-term and the long-term recovery group, with these composed individual plots at each measurement point are also shown in Fig. 3. The SP short term recovery group showed about 64% recovery of muscle mass, and about 28% recovery of tetanic tension output relative to the contralateral side (Figs. 3A and 3B). Then, the SP long term recovery group showed increased recovery of average muscle mass (77%), and significant recovery of tetanic tension output (49%, Figs. 3A and 3B). Importantly, the media control group showed significantly (P < 0.05) lower recovery at 4 week (Fig. 3A) with strong atrophy coalescence into the tibial bone (see next Figs. 4I–4K), and incomplete recover of the muscle–tendon unit continuity. This atrophy apparently progressed toward 6 weeks (compare I and J). Thus, it was clear that this model is irreversible spontaneously. Therefore, we were unable to continue measuring muscle mass and tension output in the media Control group subsequently. Average recovery ratio of muscle mass gradually increased in the long-term group, and the tetanic tension showed significantly higher recovery (Figs. 3A and 3B), showing that a development of recoveries continued/progressed over 14 weeks after operation.

Figure 3 Differences of operated TA muscle mass and tetanic tension output between the short (4–8 weeks) and long (14–18 weeks) term groups, and these composed individual plots at each measurement point.

Average recovery (%) of operated muscle mass (A) and tetanic tension output (B). In muscle mass recovery in the short term, the SP group showed significantly higher values than C group, and recovery was progressed in the long term group (but not significant) (A). Similarly, significantly higher tension recovery was observed in the long term group (B). (C and D) Individual distributions of recovery was also analyzed in muscle mass and tension. Clear linear relationships between term and recovery are detected on the individual plots of the long term group, but not in the short term group both in the muscle mass (C) and tetanic tension output (D).

When the recoveries were analyzed individually in each group, a linear relationship between the term and the manner of recoveries was observed to be relatively stronger in the long-term group than that of the short term group. This may be representing the effect of increase in standard body activities of mice in the cages, because we observed that general activities (walking around, hanging and upside-down walking on the cage lid) clearly occurred more frequently in the long-term recovery group than in the short term group.

Macroscopic examination

Typical fluorescence macroscopic features at 4–10 weeks after the engraftment of transplanted GFP+ sheet-pellets are shown in Fig. 4. In situ observation revealed that a large volume of GFP+ tissues was engrafted in the damaged TA muscle portion after transplantation, and these tissues showed 5 different patterns. Pattern 1 was the most common one (5/13 samples), showing broad and thick engraftment through the tendon to the mid portion of the muscle (Figs. 4A–4C). GFP+ engrafted tissue was observed in blood vessel networks (arrows in Fig. 4B), and the continuity was clearly maintained (Fig. 4C). This pattern was mainly composed of connective tissue with few muscle fibers. Pattern 2 was rarely observed (1/13 samples), and it mostly comprised of the muscle fibers (Fig. 4D) that extended from the mid portion of MTJ to the upper portion, and with few connective tissues. Pattern 3 (1/13 samples) in contrast, mainly comprised of the connective tissue at the distal portion of MTJ, with few muscle fibers (Fig. 4E). Patterns 4 (3/13 samples) and 5 (3/13 samples) were mixed types, showing both muscle fibers and connective tissues (Figs. 4F–4H), but GFP was relatively sparse, and the amount of connective tissues was small in Pattern 4 (Fig. 4F), while Pattern 5 showed even distribution of both tissues (Fig. 4G). Active blood vessels in and around GFP+ tissues were equally observed in all Patterns (arrows in Figs. 4B, and 4D–4H). Severe atrophy of TA muscle was generally observed in the C group (Figs. 4I and 4J), and this pattern was observed in 7/7 control mice. On the lateral side of the tibia (black arrows in Figs. 4I and 4J), a portion of the TA muscle was clearly hollowed (white arrows in Fig. 4I), and there were no muscle fibers present. Apparent adipose tissue formation (red arrows in Fig. 4J) was also observed, suggesting lack or very weak continuity of the TA muscle–tendon unit (Fig. 4J). An apparent muscle deficit of distal-half of operated TA with a meager tendon was confirmed in the isolated TA muscle of C group (K), suggesting the difficulty of force generation. Progressive muscle atrophy of the remaining proximal-half portion of TA was also apparent (compare I and J in dotted circles).

Figure 4 Macroscopic observation of surgically treated TA muscles at 5–10 weeks (W) after transplantation in situ and in vitro (after removal).

Photographs were taken by synchronizing light conditions as fluorescence + normal. (A–G) Typical features of sheet-pellet (GFP+) transplanted muscles. (A–C) Pattern 1, (D) Pattern 2, (E) Pattern 3, (F) Pattern 4, and (G–H) Pattern 5. Arrows in (B, D, E, F, G), and (H) show blood vessels. The relationship between the ingression of blood vessels and GFP+ tissues can be also confirmed in Fig. S1. (I and J) Typical features of media transplanted control muscles. White arrows in (I) show the dent in the TA position. Red arrows in (J) show fat tissue. When the media control muscle was isolated (in vitro), a deficit of distal-half of TA muscle with a meager tendon was apparent (arrows in K). The remaining proximal-half of TA (dotted circles in I–K) showed progressive muscle atrophy toward the 6 weeks (compare I and J). Black arrows in (I) and (J) indicate tibial bone. Bars = 1 mm.

Figure 5 Immunohistochemical detection of engrafted Sk-MSCs in cross-section (operated muscle from Pattern 1, 7 weeks after operation).

(A and B) Relationship among the GFP+ cell-derived connective tissue, recipient tendon, and muscle fibers. Skeletal muscle fibers were stained with Sk-actin (skeletal muscle actin, A), and tendon and connective tissues were stained with Elastica Van Gieson (B). (C) Endothelial cell staining with CD31. White arrows show GFP+/CD31+ cells. Yellow arrows show penetration of GFP+ cell-derived connective tissue into the tendon. Dotted line drawing shows the contours of the tendon. (D) Axon staining with Neurofilament 200 (N200). (E) Schwann cell staining with p75. (F) Dystrophin staining for skeletal muscle fibers in the MTJ. Muscle fibers expressing dystrophin (red reactions). Dotted line squares in (A) representing small letter c, d, and e correspond with the portions of which the (C-E) was obtained. Higher magnification photographs, showing the GFP+/CD31+ cells and/or GFP+/SMA (anti-smooth muscle actin)+ cells, were also available in Fig. S2. Similarly, localization of GFP+/p75+ cells can be also confirmed in Fig. S3 on the website. T, tendon. Dotted lines in (D) and (E) show the border between connective tissue and muscle fibers. Blue staining, DAPI. Bars in (A) and (B) 200 µm, (C–E) 100 µm.

Immunohistochemical analysis of the engrafted cells

At 4–10 weeks following the surgery, the behavior of engrafted GFP+ cells was analyzed in cross-sections. Figure 5 shows the result of Pattern 1. Thick tissues composed of GFP+ cells closely adhered to skeletal muscle fibers (Fig. 5A). GFP+ cells surrounded the tendon, and van Gieson elastic fiber staining showed connective tissue networks (Fig. 5B). This suggests that the engrafted GFP+ tissue formed connective tissue networks, which connected both the tendon and the muscle fibers, displaying a “bio-bond” role. Additionally, invagination of GFP+ cell-derived connective tissue was observed in the tendon (yellow arrows in Fig. 5C), and showed differentiation into vascular endothelial cells (CD31 + GFP, white arrows in 5C). A similar trend was also observed on the muscle fiber side, and the migration of GFP+ cells could be seen between muscle fibers (Fig. 5D, below the dotted line). A close relationship (not double staining) of N200+ nerve axons and GFP+ cells, was observed on both the muscle side (below the dotted line) and the connective tissue side (Fig. 5D, upper-side of dotted line). GFP+ cells in the connective tissue were positive for p75 (Fig. 5E, right side of dotted line) and therefore, they were considered Schwann cells. These results support a close relationship between GFP+ cells and nerve axons, which is detectable in Fig. 5D. The relationship among engrafted GFP+ connective tissue, tendon, and muscle fibers is more apparent in Fig. 4F. GFP+ connective tissue bridging tendon (T) and muscle fibers (right side of the panel) expressed dystrophin (red), which suggests a strong relationship between connective tissue and muscle fibers through dystrophin complexes.

Similarly, longitudinal profiles obtained from pattern 5 are shown in Fig. 6. Engrafted GFP+ cell-derived connective tissues closely adhered to the muscle fibers (Fig. 6A), and GFP+ muscle fibers were also observed (Fig. 6B). Differentiation of GFP+ cells into vascular smooth muscle cells was seen in the connective tissue network (Fig. 6C, arrows, double staining with GFP + SMA), contributing to a relatively large blood vessel formation. The involvement in the peripheral nerve reconstitution was indicated by a close relationship between GFP+ cells and axons (Fig. 6D, arrows, close distributions of GFP+ cells and N200+ axons). This relationship was supported by the differentiation of GFP+ cells into Schwann cells positive for p75 (Fig. 6E, arrows, double staining of GFP + p75). Similarly, a close relationship of GFP+ cells and muscle fibers, nerve axons, and the neuromuscular junctions was evident (Fig. 6F, in the dotted line circle as α-bungarotoxin+), confirming that GFP+ cells contributed to peripheral nerve extensions, reaching to the end of a motor nerve. These results indicate that the transplanted GFP+ Sk-MSC sheet-pellets mainly form connective tissue networks together with a certain amount of muscle fibers, and that they physically bind the tendon and muscle fibers, contributing to the peripheral nerve-blood vessel formation.

Figure 6 Immunohistochemical detection of engrafted Sk-MSCs in longitudinal sections (operated muscle from Pattern 5, 8 weeks after operation).

(A) Close relationship between donor-derived GFP+ connective tissue and muscle fibers was also apparent in the longitudinal images. (B) GFP+ muscle fibers were observed. (C) Vascular smooth muscle staining. (D) Axon staining by N200. (E) Schwann cell staining by p75. (F) Axon and neuromuscular junction staining by N200 and α-Bungarotoxin. Dotted circle in (F) shows the position of neuromuscular junction. Mf, muscle fiber. Blue staining, DAPI. Bars = 100 µm.

Discussion

Injuries involving the muscle belly or MTJ are challenging for surgeons because the muscle tissue shows poor suture-holding capacity, and the reliable suture methods have not been established yet (Faibisoff & Daniel, 1981; Kragh et al., 2005c; Oliva et al., 2013; Phillips & Heggers, 1988). However, reconstruction and/or re-establishment of continuity in the muscle–tendon unit is vital for the functional repair, because of the primary role of muscle force generation and transmission. We have investigated the reconstruction of completely ruptured TA muscle at the MTJ, using Sk-MSC sheet-pellets as “bio-bonds”. The results indicate that Sk-MSC sheet-pellet transplantation achieved favorable results in the reconstruction and/or reconnection of the ruptured muscles and tendons. Engrafted Sk-MSCs primarily formed connective tissues, including neurovascular networks, and bridged both the tendon and muscle fibers, with differentiation into skeletal muscle fibers, Schwann cells, vascular smooth muscles, and endothelial cells. The differentiation capacity of these cells was previously predicted, since the present sheet-pellet was mainly composed of the mixed population of multipotent Sk-34 (CD34+/45−) (Tamaki et al., 2002; Tamaki et al., 2005) and Sk-DN (CD34−/45−) (Tamaki et al., 2003; Tamaki et al., 2007a; Tamaki et al., 2007b) cells, and their putative potential for the therapy was confirmed by RT-PCR and protein array analysis immediately before the transplantation (Figs. 2A and 2B). Furthermore, the migration of engrafted GFP+ fibroblast-like cells was observed around the tendon and the interstitium of muscle fibers, mechanically bridging the tendon-muscle gap. This reconnecting behavior is considered the “bio-bond” activity. Establishment of a muscle fiber holding capacity in donor-derived connective tissue was further suggested by the expression of dystrophin in MTJ fibers (Fig. 5F), because of the role of dystrophin-dystroglycan (α, β) complex in the collagen network (Monti et al., 1999; Welser et al., 2009). In our previous studies, we prepared the sheet-pellets as the accelerators of the neurovascular reconstitution with diminished myogenic potential (Soeda et al., 2013; Tamaki et al., 2013). Here, we used a shorter term expansion culture (in particular, the term of first fiber culture), and this shorter culture period helped preserve the myogenic potential of the sheet-pellets, resulting in new myofiber formation. This was confirmed by the comparison of the RT-PCR data obtained in the previous studies (Soeda et al., 2013; Tamaki et al., 2013) and the results obtained in this one (Fig. 2). The relatively preserved myofiber formation capacity may have also contributed to the enhanced connection of the muscle–tendon unit.

The investigated sheet-pellets contributed to functional regeneration of disrupted TA muscles by 36% in the short term recovery group, and this increased 49% in the long term recovery group. Concerning to the relationship between the engrafted patterns (Fig. 4) and functions, relatively higher contractions were dominantly observed in the Pattern 4 and 5 (Figs. 4F and 4G), which showed mixed types of GFP engraftment of both muscle fibers and connective tissues. The tension recoveries were further supported by the immunohistochemical staining results, showing that GFP+ cells contributed to motor nerve extensions, close to the neuromuscular junctions (Fig. 6F, from Pattern 5). We did not perform the tensile strength test of the regenerated muscles, but no muscle ruptures were observed during repetitive maximum tetanic tension measurements. In addition, the absolute value of the tension output of operated-muscles (around 38 g) was superior to the mean body weight (around 34 g) in the long term group. Therefore, we believe that the tensile strength of operated-muscle may have been sufficient to support standard body activities, and it could be enhanced, as one type of rehabilitation, which resulted in an increased tension recovery in the long term group (Table 1 and Figs. 3A and 3B). The notion that rehabilitation (increased body activity) depends on an increase in the tension recovery was further suggested by the term-dependent linear relationship, as was observed in the individual plots of the long term group (Fig. 3D). We observed greater general cage activities in the mice in the long term group compared to the short term group, and the same trend was also detected in the muscle mass recovery (Fig. 3C). Thus, we believe that this linear relationship may be due to the increased general activity of mice day by day following the improvement in the general symptoms, such as reduced pain and discomfort. In this regard, it was suggested that these symptoms were more prominent in the short term group, and may induce lower activities. However, this lower activity might be beneficial for tissue recovery. For these reasons, it was also suggested that starting rehabilitation earlier, probably around 8 weeks after the operation, may decrease recovery time and improve recovery itself.

As a consequence, the non-transplanted control group did not achieve sufficient reconnection of muscle–tendon units (Fig. 4 and Table 1), showing that the MTJ rupture model we used is an irreversible model. It has been reported that the early reconnection induced mechanical stimulation is effective for the repair and quality maintenance of the tendon, because the lack of this stimulation produced detrimental effects (Lin, Cardenas & Soslowsky, 2004; Matsumoto et al., 2003). Therefore, we concluded that the absence of mechanical-tension in the control group, after the complete removal of the MTJ, may have caused a massive muscle atrophy associated with the fatty tissue replacement (Fig. 4). It is also a fact that the present sheet-pellets transplantation prevented these detrimental effects.

Additionally, it has been suggested that early peripheral nerve regeneration and the provision of neuropeptides are important for the healing of normal connective tissue and tendon (Ackermann, Ahmed & Kreicbergs, 2002; Ackermann et al., 2003). Neovascularization plays a critical role in the healing process of the ligament (Nishimori et al., 2012). The repair of angiokinesis (vasodilation-constriction) through the neurovascular regeneration is also important for tissue regeneration (Ackermann, Ahmed & Kreicbergs, 2002). Our treatment meets all these conditions. Previous studies demonstrated that Sk-MSCs transplantation facilitates/accelerates nerve-vascular formation (Tamaki et al., 2014; Tamaki et al., 2007a; Tamaki et al., 2013; Tamaki et al., 2005). The expressions of various nerve-blood vessel-related growth and trophic factors before and after transplantation was also observed in the case of nerve-gap regeneration (Tamaki et al., 2014). In our study, prolonged expression of these genes, shown in Fig. 2, was expected and observed following the transplantation, and it may be beneficial for the healing of muscle–tendon units.

We recently established a practical/therapeutic method of isolation of human skeletal muscle-derived stem cells (Tamaki et al., 2015). Using this method, we found that human cells can be divided into 2 stem/progenitor cell fractions; (1) the cells showing preferential differentiation into the skeletal myogenic lineage (CD45−∕CD34−∕29+ = Sk − DN∕29+), and (2) cells showing multiple differentiation into nerve-blood vessel cell lineages (CD45−∕34+ = Sk − 34). The combined differentiation/reconstitution capacities of these cells after in vivo transplantation were comparable to the mouse Sk-MSCs (Tamaki et al., 2015). Therefore, the cell fractions could be selected and adjusted for the treatment of muscle fibers, nerve-blood vessels associated with connective tissues, or both. Additionally, these stem/progenitor cells can be obtained from various muscle regions, including the legs and abdominals (Tamaki et al., 2015). The removal of a small sample (around 3 g) from the lower abdominal wall muscle carries low risk to the loss of motor function.

Conclusions

In the present study, we successfully demonstrated that Sk-MSC sheet-pellet transplantation can bridge complete rupture of MTJ, and form connective tissue networks associated with cellular differentiation into skeletal muscle fibers, Schwann cells, vascular smooth muscle, and endothelial cells. These connective tissues migrated around the tendon and muscle fiber interstitium and connected both tissues, playing a role of a “bio-bond.” Paracrine effects of nerve and vascular growth factors, as well as trophic factors, produced by Sk-MSCs may also be beneficial for the reparation of the tissue. Together with the recent establishment of human skeletal muscle-derived cells, Sk-MSCs may be an optimal autologous cell source, used as an adjuvant, which would lead to a promising therapy for muscle–tendon injuries. Furthermore, the combined therapies, such as an appropriate suture with sheet-pellet, or suture together with sheet-pellet and scaffold use, may also represent favorable therapies for the MTJ rupture repair, particularly because of the higher tensile strength required in humans.

Supplemental Information

Data S1 Tension raw data, and primary photos of RT-PCR and protein array

Click here for additional data file.

Table S1 Specific primers for mice

Specific primers and the analyzed materials are summarized.

Click here for additional data file.

Figure S1 High magnification photographs of the portions indicating by the arrows in Figs. 4B, 4D, 4E and 4F

Interactions between blood vessels and GFP+ tissue are evident.

Click here for additional data file.

Figure S2 Higher magnification photographs, showing the GFP+/CD31+ cells and/or GFP+/SMA+ (anti-smooth muscle actin) cells

These are evidences of the differentiation of GFP+ donor cells into vascular endothelial cells (CD31) and vascular smooth muscle cells (SMA).

Click here for additional data file.

Figure S3 Higher magnification photographs showing the localization of GFP+/p75+ cells

These are evidence of the differentiation of GFP+ donor cells into Schwann cells.

Click here for additional data file.

Additional Information and Declarations

Competing Interests

Author Contributions

Animal Ethics

Data Availability

The authors declare there are no competing interests.

Hiroyuki Hashimoto conceived and designed the experiments, performed the experiments, analyzed the data, prepared figures and/or tables.

Tetsuro Tamaki conceived and designed the experiments, performed the experiments, analyzed the data, contributed reagents/materials/analysis tools, wrote the paper, prepared figures and/or tables.

Maki Hirata performed the experiments, analyzed the data, contributed reagents/materials/analysis tools, prepared figures and/or tables.

Yoshiyasu Uchiyama conceived and designed the experiments, analyzed the data.

Masato Sato reviewed drafts of the paper, overview.

Joji Mochida reviewed drafts of the paper, research funding, overview.

The following information was supplied relating to ethical approvals (i.e., approving body and any reference numbers):

All experimental procedures were approved by the Tokai University School of Medicine Committee on Animal Care and Use (153015).

The following information was supplied regarding data availability:

The raw data has been supplied as a Supplemental Dataset.

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
