# Peer review of "Reconstitution of the complete rupture in musculotendinous junction using skeletal muscle-derived multipotent stem cell sheet-pellets as a “bio-bond”"

_PeerJ, doi:10.7717/peerj.2231_

## Round 0.1 · original submission · Major Revisions

I first apologize for the length of time it has taken to return my decision to you. It was very difficult to obtain reviewers for this manuscript.

Two reviews found the paper a worthwhile contribution, indicate minor revision and overall liked the paper. Our first reviewer identifies problems that need to be taken into account. I concur with this reviewer in that the study design seems to be rather poor, and not enough data to draw your conclusions are provided. Macroscopic pictures of the sheet pellets before transplantation should be added. In the immunohistochemical analysis, false positive results are not prevented. Reviewer two also pointed out problems with the contours of the sheet pellet, which even lack of concept definition in the introduction section (see our third review). Two of the reviewers found your figures problematic (for example 3 and 5), please take these opinions in full consideration.

I would like you to follow all the suggestions or our three reviewers.

Reviewer 1 ·

Basic reporting

All my comments are described in “General Comments for the Author.”

Experimental design

All my comments are described in “General Comments for the Author.”

Validity of the findings

All my comments are described in “General Comments for the Author.”

Additional comments

General
The idea for skeletal muscle-derived multipotent stem cell sheet-pellets as a “biobond” is interesting, however, the study design seems to be poor, therefore, the authors could not provide enough data to draw their conclusions.

Specifics

Macroscopic pictures of the sheet pellets before transplantation should be demonstrated.

For example, it seems to be important to compare muscle mass (mg) of Op-side in the SP group and muscle mass (mg) of Op-side in the C group at 4 weeks. If muscle mass (mg) of Op-side in the C group at 4 weeks was unmeasurable, another assay (for example, cross section area) should have been performed. The situation is similar for tetanic tension.

4w (n=1), 5w (n=1), 6w (n=1), 8w (n=1), 14w (n=1), 15w (n=1), 16w (n=1), 18w (n=1); these sample numbers seems to be strange.

Also, Figure 3C and 3D seem to be strange.

For immunohistochemical analysis, false positive for GFP is always problematic. In Figure 5 and 6, most cells appeared to be GFP positive. False positive for GFP cannot be denied.

The article structure should be modified. Many discussions are described in "Results".

·

Basic reporting

1- Table 1 should be indicated as supplementary data
2- Results in table 2 are the same as those shown as a histogram in figures 3a and 3b. This redundancy is not useful.
3- In figure 1-4, it would be suitable to define the contours of the sheet pellet .
4- Blood vessels showed in figure 4 by white arrows are not really visible, especially in photographs B and D. It would be suitable to add an insert showing blood vessels with a higher magnification.
5- Line 644: add (W) after 5-10 weeks
6- Figures 5c are too low magnification. An insert showing endothelial cells enclosed to GFP+ cells at higher magnification would improve the readability of the figure. Similarly, it would be clearer to define the contours of the tendon in figure 5c.
7- The meaning of yellow arrows is not indicated in the legend of figure 5.

Experimental design

No comments

Validity of the findings

No comments

Reviewer 3 ·

Basic reporting

The background literature review is sufficient for the purpose. However, the literature review on the literature most relevant to this innovation (lines 75 to 89, especially 86-89) is rather brief considering the novelty of what is proposed. A more detailed review demonstrating exactly where the gaps are in the relevant literature, and the precise aspects of the novelty of this work is recommended. Is the work cited in lines 86 to 89 really the only literature exploring stem cell solutions to of muscle-tendon injuries? I am well aware the manuscript has a large number of references already, but nothing could be more important than to address the issues I mentioned above, and if some less important references needed to be deleted to accommodate this, then it is justified, although in my opinion the addition of extra references should not necessitate the deletion of others.

Specific amendments required:

1. The important but unclear concept of the “sheet-pellet” needs to be precisely defined in the introduction. A pellet is generally an equi-axed geometry, such as a small equi-axed cylinder or small sphere, while a sheet has planar geometry. How are the stem-cell blends incorporated therapeutically into this conflicting geometric descriptor?
2. Authors need to add further text either clarifying that the work cited in lines 86 to 89 is really the only literature exploring stem cell solutions to of muscle-tendon injuries, or alternatively devote more of the literature review reviewing literature on this specific and important issue not currently included in the paper.

Experimental design

Methodology and data reporting is appropriate and adequate.

Validity of the findings

The results appear to be scientifically valid and present a convincing case for the effectiveness of this therapy. The authors are to be commended on good experimental design, and their development and validation of a successful treatment innovation.

Additional comments

Specific amendments required:

1. The important but unclear concept of the “sheet-pellet” needs to be precisely defined in the introduction. A pellet is generally an equi-axed geometry, such as a small equi-axed cylinder or small sphere, while a sheet has planar geometry. How are the stem-cell blends incorporated therapeutically into this conflicting geometric descriptor?
2. Authors need to add further text either clarifying that the work cited in lines 86 to 89 is really the only literature exploring stem cell solutions to of muscle-tendon injuries, or alternatively devote more of the literature review reviewing literature on this specific and important issue not currently included in the paper.

---

## Round 0.2 · Minor Revisions

I am ready to accept your paper, I have however a last request. I am unable to distinguish the cells that the yellow arrows are pointing out in your figure 5C (first and second boxes). Could you please totake account of this?

---

## Round 0.3 · accepted · Accept

Nice work, I am happy to accept it